# Comparative Genomic Analysis of Cold-Water Coral-Derived *Sulfitobacter faviae*: Insights into Their Habitat Adaptation and Metabolism

**DOI:** 10.3390/md21050309

**Published:** 2023-05-19

**Authors:** Shituan Lin, Yunxue Guo, Zixian Huang, Kaihao Tang, Xiaoxue Wang

**Affiliations:** 1Key Laboratory of Tropical Marine Bioresources and Ecology, Guangdong Key Laboratory of Marine Materia Medica, Innovation Academy of South China Sea Ecology and Environmental Engineering, South China Sea Institute of Oceanology, Chinese Academy of Sciences, Guangzhou 511458, China; linshituan19@mails.ucas.ac.cn (S.L.); yunxueguo@scsio.ac.cn (Y.G.); huangzixian21@mails.ucas.ac.cn (Z.H.); khtang@scsio.ac.cn (K.T.); 2University of Chinese Academy of Sciences, Beijing 100049, China; 3Southern Marine Science and Engineering Guangdong Laboratory (Guangzhou), Guangzhou 511458, China

**Keywords:** *Sulfitobacter*, cold-water coral, prophages, genomic islands, toxin-antitoxin system

## Abstract

*Sulfitobacter* is one of the major sulfite-oxidizing alphaproteobacterial groups and is often associated with marine algae and corals. Their association with the eukaryotic host cell may have important ecological contexts due to their complex lifestyle and metabolism. However, the role of *Sulfitobacter* in cold-water corals remains largely unexplored. In this study, we explored the metabolism and mobile genetic elements (MGEs) in two closely related *Sulfitobacter faviae* strains isolated from cold-water black corals at a depth of ~1000 m by comparative genomic analysis. The two strains shared high sequence similarity in chromosomes, including two megaplasmids and two prophages, while both contained several distinct MGEs, including prophages and megaplasmids. Additionally, several toxin-antitoxin systems and other types of antiphage elements were also identified in both strains, potentially helping *Sulfitobacter faviae* overcome the threat of diverse lytic phages. Furthermore, the two strains shared similar secondary metabolite biosynthetic gene clusters and genes involved in dimethylsulfoniopropionate (DMSP) degradation pathways. Our results provide insight into the adaptive strategy of *Sulfitobacter* strains to thrive in ecological niches such as cold-water corals at the genomic level.

## 1. Introduction

The genus *Sulfitobacter* is one of the major sulfite-oxidizing alphaproteobacterial groups in aquatic environments and belongs to the family *Rhodobacteraceae* [1]. The first strain of this genus, *Sulfitobacter pontiacus*, was isolated from the H_2_S-O_2_ interface of the Black Sea, and the genus was proposed by Sorokin in 1995 [2]. Later, other strains of this genus were isolated from a diverse range of habitats, including tidal flat sediments, deep seawater, and hypersaline lakes [3,4]. Species of the genus *Sulfitobacter* are chemolithoheterotrophic and have the ability to obtain additional energy from sulfite oxidation in acetate-limited aerobic conditions [2]. They are Gram-negative, ovoid or rod-shaped, and mostly catalase- and oxidase-positive [1].

Species of the *Sulfitobacter* genus are often associated with marine algae and corals in shallow waters [3,4,5,6] and were also found across corals, including the deep-water coral taxa *Lophelia pertusa*, *Madrepora oculata*, and *Paragorgia arborea* [7]. Although sulfur cycling in corals and their symbionts has not been well investigated, sulfur-containing compounds have been detected in coral soft tissues and skeletons [8]. Previous work suggests that sulfide-oxidizing bacteria could supply organic matter to the associated benthic marine animals; for example, sulfide-oxidizing symbiotic bacteria produced oragnic matter that is exploited by the vestimentiferan tube worms in the hydrothermal vents [9]. Additionally, the resultant sulfate is incorporated into the tissues of the reef-building coral *Acropora tenuis* collected around Sesoko Island in Okinawa and converted to other sulfur compounds such as sulfated glycosaminoglycans and lipids [8]. Furthermore, dimethylsulfoniopropionate (DMSP) is abundant in the oceans, and the catabolism of DMSP is an important step of the sulfur cycle [10]. DMSPs are produced by phytoplankton, macroalgae, heterotrophic bacteria, and corals [4,11,12,13]. Studies have shown that DMSP can help coral alleviate intracellular oxidative stress, which is considered important to mitigate coral bleaching or death in tropical scleractinian corals [14,15]. It has also been proposed that DMSP plays a key role in structuring coral-associated bacterial communities, and a correlation was observed between DMSP availability and the DMSP demethylase DmdA-positive microbes, including *Sulfitobacter*, in some coral species [16]. In addition, DMSP produced by coral or algal hosts may also affect the lifestyle of *Sulfitobacter*. A recent study showed that the *Sulfitobacter* D7 strain undergoes a lifestyle switch from coexistence with the host to induction of host death, and DMSP produced by the algal host is a key chemical component that triggers the switch in lifestyle of *Sulfitobacter* D7 [17].

Species of the *Sulfitobacter* genus are widely distributed across different geographical regions, depths, and hosts, and their genomes vary both at the interspecies and intraspecies levels. Mobile genetic elements (MGEs) are known to act as a major force for bacterial genetic variation and adaptation to a diverse range of habitats, including the human stomach and the hot spring [18,19,20,21]. In this study, we isolated two strains belonging to *Sulfitobacter faviae* from two different cold-water black corals, *Dendrobathypathes* sp. and *Telopathes* sp., at depths of ~1000 m in the South China Sea. To explore the differences in MGEs between the two strains, a comprehensive comparative genomic analysis was conducted. We found that even though their chromosomes shared high similarity, the MGEs, including plasmids, genomic islands, and prophages, differed greatly. These MGEs carry multiple genes that are involved in important biological processes, such as phage defense and stress responses, which may benefit the bacteria in deep-sea environments. In addition, the two strains shared similar sulfite oxidation and DMSP degradation enzymes and secondary metabolite biosynthetic gene clusters. Thus, this study provides insight into the adaptive strategy of *Sulfitobacter* strains to thrive in ecological niches such as cold-water corals.

## 2. Results and Discussion

### 2.1. The Taxonomic Status of Sulfitobacter faviae Strains SCSIO W1865 and SCSIO W1866

Two cultures, SCSIO W1865 (isolated from black coral *Dendrobathypathes* sp. at the depth of 1059 m) and SCSIO W1866 (isolated from black coral *Telopathes* sp. at the depth of 1072 m), were observed under the transmission electron microscope (TEM). Both SCSIO 1865 and SCSIO 1866 were ovoid or rod-shaped (approximately 0.7–2 µm in diameter) bacteria with long single polar flagella (~4–5 µm in length), which may help them swim in liquid environments (Figure 1A). Sequences of 16S rRNA genes from the two cultures showed ~99% similarity to *Sulfitobacter faviae* S5-53^T^ isolated from the brain coral *Favia veroni* in the Andaman Sea, India. Phylogenetic analysis of both 16S rRNA genes (Figure 1B) and the whole genome (Appendix A) with 22 other *Sulfitobacter* spp. also showed that SCSIO W1865 and SCSIO W1866 fell in the clade with *Sulfitobacter faviae*. Further analysis based on average nucleotide identity (ANI) across whole genome sequences showed that SCSIO W1865 and SCSIO W1866 shared 95.98% and 95.79% ANI values with *S. faviae* S5-53^T^, respectively, suggesting that SCSIO W1865 and SCSIO W1866 both belong to *S. faviae*. Thus, the two strains were identified as *Sulfitobacter faviae* SCSIO W1865 and SCSIO W1866 (Figure 1C).

### 2.2. Genome Features and Annotations of Sulfitobacter faviae Strains SCSIO W1865 and SCSIO W1866

The genomes of the two strains were sequenced with an Illumina and PacBio hybrid strategy. One circular chromosome of 3.24 Mbp and four circular plasmids of 287.79 kbp, 209.19 kbp, 182.75 kbp, and 103.48 kbp in size were recovered in SCSIO W1865 (Table 1). Similarly, one circular chromosome of 3.15 Mbp and three plasmids with 210.70 kbp, 146.56 kbp, and 100.91 kbp were also recovered in SCSIO W1866. The G+C contents of the two strains are ~60%. SCSIO W1865 and SCSIO W1866 chromosomes encode 3185 and 3122 coding DNA sequences (CDSs), respectively. Both chromosomes encode 49 tRNA genes and 12 rRNA genes, while none of the plasmids encode tRNA or rRNA, indicating that the megaplasmids rely on host tRNA and rRNA pools to translate.

Overall, the two chromosomes were highly similar (96.26% identity) and shared 3101 core genes (Figure 2). SCSIO W1865 and SCSIO W1866 each encoded 738 and 397 strain-specific genes, respectively (Figure 2). The genes involved in lipid metabolism that may allow them to sustain life at low temperatures are listed in Appendix A. In addition, SCSIO W1865 and SCSIO W1866 both encode antifreeze proteins that function in cold adaptation [22]. Among these strain-specific genes, 57% and 35% were located in endogenous megaplasmids in SCSIO W1865 and SCSIO W1866, respectively. In addition, 18% and 41% of them were located in other MGEs (prophages, genomic islands, et al.), accordingly. Altogether, ~75% of the strain-specific genes were harbored by MGEs, suggesting that these MGEs contribute to the genetic diversity between the two strains.

### 2.3. Prediction of Mobile Genetic Elements (MGEs)

#### 2.3.1. Plasmids

The four megaplasmids in SCSIO W1865 encoded 276, 192, 188, and 98 CDSs, and the three megaplasmids in SCSIO W1866 encoded 195, 152, and 116 CDSs (Table 1). Two plasmids were highly similar in the two strains (~96% identity), and the rest were unique (Figure 3 and Appendix A). Functional analysis of the genes encoded by the megaplasmids in each strain was performed, and similar GO (Gene Ontology) pathways were annotated, including metabolic processes related to phosphorus, carbohydrate, nitrogen compounds, and urea; responses to stimulation including ion, osmotic stress, and oxidative stress; and some other pathways, especially those involved in defense mechanisms and motility (Appendix A). SCSIO W1865 megaplasmids encoded genes with specific molecular functions in ATP-dependent activity and molecular transducer activity (Appendix A). These results suggested that the megaplasmids in the two strains regulated physiological processes and competition with other bacteria in the environment through different genes that are involved in similar important pathways.

#### 2.3.2. Prophages

Integrated prophages can affect or alter bacterial host conditions by prophage induction, prophage genome excision, or prophage gene expression. For example, prophage excision acts as a regulatory switch to enable the survival of the *Shewanella oneidensis* host at low temperatures [21]. We have found that activation of prophages plays a major role in mediating colonization competition in the shallow-water scleractinian coral *Galaxea fascicularis* microbiota [23]. In addition, individual prophage genes that are expressed in a silent prophage condition can increase host fitness, especially during conditions of stress, including oxidative and nutrient stresses [24].

Here, we searched for prophage candidates in both the SCSIO W1865 and SCSIO W1866 genomes using PHASTER [25]. In SCSIO W1865, two prophages were found, Prophage 1 of 55.8 kb and Prophage 2 of 17.5 kb, while in SCSIO W1866, only one prophage (Prophage 3) was found that shares a high level of similarity with Prophage 2 (96.03%) in SCSIO W1865 (Figure 4). By searching phage genomes in the NCBI virus database, no similar phage sequences were obtained for prophages 1–3, suggesting that they may be prophages that were not identified previously. To further validate this, the major capsid proteins (MCPs) and terminases were also searched, and they all showed similarities (~50% identity) to those of Myoviridae phages. Prophage 1 was integrated downstream of the tRNA^ser^-encoding gene with two identical attachment sites, 5′-CGCGCGCCACCGCC-3′, and the insertion of Prophage 1 generated a gene encoding dTDP-glucose 4,6-dehydratase; thus, the excision of prophage 1 will lead to the disruption of the coding region of this gene. Prophages 2 and 3 share high similarity and were inserted at the same sites, the intergenic region between murein endolytic transglycosylase MltG and the pyruvate dehydrogenase E1 component alpha subunit, but no known integrase was identified.

In addition to phage structural or replication-related genes, prophage 1 encodes an RNA polymerase sigma factor that can combine with or compete with the host RNA polymerase sigma factor to selectively regulate the transcription of certain genes by recognizing specific promoters. Prophages 2 and 3 encode general stress proteins and proteases, and they may help hosts deal with environmental stresses and the production of certain harmful proteins. Prophages are activated by environmental stress, including temperature, UV, oxidative stresses, and nutrient limitation [26], and activation of these prophages may contribute to physiological modification of their hosts and the interaction among competitive bacteria, which in turn could affect the health of deep-sea cold-water coral holobionts. The function of the two prophages in SCSIO W1865 and SCSIO W1866 remains to be determined.

#### 2.3.3. Genomic Islands (GIs)

We have reported that mobile GI mediates the competition between coral microbial populations [27]. Here, we predicted the presence of potential GIs on both chromosomes using IslandViewer 4 [28]. As a result, eight distinct GIs were predicted in SCSIO W1865 (GI 1–8) and eleven distinct GIs were predicted in SCSIO W1866 (GI 9–19), respectively (Figure 5). These GIs encoded a series of functional genes, including ATP binding protein (GI 1, GI 7), kinase (GI 2, GI 5, GI 8), AAA family ATPase (GI 3), ssDNA specific exonuclease RecJ (GI 7), DNA methyltransferase (GI 7), N-acetyltransferase (GI 8), reverse transcriptase (GI 8), HNH endonuclease (GI 8, GI 9), transcriptional regulators (GI 2, GI 4 to 6, GI 9, GI 12 to 14, GI 16 to 18), RNA polymerase subunits (GI 10), response related genes (GI 11, GI 16), lysozyme (GI 14), anti-sigma regulatory factor (GI 15), Lon protease (GI 18), elongation factor Tu (GI 19), and DNA topoisomerase IV subunit (GI 19). These genes can be involved in host physiology at the transcriptional, translational, and posttranslational levels. In addition, some GIs encode their own integrases/recombinases/transposases, which may mediate their mobility among bacteria and thus affect their physiological and ecological roles.

### 2.4. Antiphage Systems

The interplay between phages and bacteria is central to the ecology and evolution of microbial communities. Bacteria have evolved numerous antiphage systems to prevent phage attack and lysis, including CRISPR–Cas systems, TA systems, restriction modification systems, retrons, and so on [29,30]. These systems interfere with the infection of phages at different stages of the phage life cycle through direct or indirect interaction with phage proteins [31,32]. The identification of antiphage systems and phage lysogenization regulators was a feature of the microbiome of coral and sponge [33,34,35], suggesting the presence of antiphage systems should play important roles in maintaining the prokaryote-marine invertebrate symbioses.

#### 2.4.1. Toxin-Antitoxin Systems

TA systems are widely distributed among bacteria and archaea, and TA systems are enriched in MGEs. Among the eight types of TA systems, Type II Tas are the most extensively studied [31,36,37], in which the toxin and antitoxin are proteins and the antitoxin neutralizes the toxicity of the toxin by direct protein–protein interaction [38]. Here, we predicted a series of TA systems belonging to type II TA systems in both SCSIO W1865 and SCSIO W1866. The majority of them were located in megaplasmids (Table 2). The megaplasmid 1 in SCSIO W1865 harbored typical HipAB and VapBC TA systems. Unexpectedly, it also harbored a new organization of the putative TA system in which the toxin is an HNH endonuclease (Dnase) [39] and the antitoxin is the typical antitoxin HigA of the TA system, HigBA. The toxin HigB in the typical HigBA TA system is an mRNase [40]. In addition, megaplasmid 3 harbored RelBE and MazEF TA systems, while megaplasmid 4 harbored another RelBE that shared no significant similarity with the one in megaplasmid 3. In SCSIO W1866, megaplasmids 2 and 3 harbored ParDE and RelBE TA systems, and this RelBE shared 100% similarity with the one in megaplasmid 3 of SCSIO W1865. In addition, the Doc/PhD TA system was also located on both chromosomes (sharing 80–90% identities). We also found that both chromosomes carry similar orphan toxins (e.g., RatA) and antitoxins (e.g., SdhE), since their counter antitoxins or toxins were not found or annotated. Because TA systems are usually small protein-coding genes, toxins or antitoxins are frequently omitted by open reading frame (ORF) prediction tools. We further confirmed that no potential toxin or antitoxin-coding genes were omitted from the intergenic region upstream and downstream of orphan toxins or antitoxins by searching potential ORFs considering six reading frames. This suggests that the two strains from deep-sea-derived cold-water corals do not obtain all the components of certain types of TA systems that can be delivered by horizontal gene transfer by MGEs. Thus, the TA systems encoded by megaplasmids can be transferred and distributed among bacteria living in the same environment [41]. TA systems are known to be involved in biofilm formation, environmental adaptation, virulence, and antiphage action [42,43], and these TA systems are also able to shape the physiological and ecological roles of bacterial hosts in marine ecosystems. For example, the ParDE family TA system ParEso/CopAso and HipAB TA system stabilize the cold adaptation-related prophage CP4So and genomic island CGI48, respectively [44,45].

#### 2.4.2. Other Antiphage Systems

In addition to TA systems, other types of antiphage systems have been identified in recent years [30,46]. Here, we analyzed the presence of these antiphage systems by combining the online database DefenseFinder and the prokaryotic antiviral defense locator (PADLOC). The structures of the predicted systems were mapped (Table 3). Both SCSIO W1865 and SCSIO W1866 harbored different antiphage systems, and those in SCSIO W1865 were distributed in chromosomes and megaplasmids 3 and 4, while SCSIO W1866 only carried antiphage systems in the chromosome. The retron antiphage system was found on the chromosome of SCSIO W1865. Although a pair of retrons was proven to be a tripartite TA system [47], most retrons have not been validated. Retrons can defend against a vast number of phages through an abortive infection mechanism [48]. As the first antiphage systems of bacteria, RM systems (including type I and type II) [49] were identified in SCSIO W1865 and SCSIO W1866 chromosomes, but their gene organizations were not totally the same, suggesting that both strains used different RM systems to escape phage infection. In addition, the SCSIO W1865 chromosome also harbored PARIS [50] and spetu systems. The antiphage systems iteAS, Shango, AbiEii, and PD-T7-2 were also harbored in megaplasmids 3 and 4 in SCSIO W1865, and they may also be exchanged among strains to arm the new hosts to fight against phages [51]. In addition to the above antiphage systems, we also predicted the most popular antiphage system, CRISPR, in the two strains with CRISPRone, and incomplete clusters of Type I CRISPR with more than 96% similarity were identified. They were both composed of five genes, and four of them were Cas genes. The one in SCSIO W1865 was located on megaplasmid 2, and the one in SCSIO W1866 was located on megaplasmid 1. The two clusters are both 3,286 bp in length (Appendix A). However, no arrays were identified. This organization suggested that the CRISPR-associated genes in the two strains may be horizontally transferred with plasmids and offer new bacteria certain physiological or ecological functions, including antiphage and gene editing.

### 2.5. Proposed Metabolism Pathways

Metabolism is essential for the habitat adaptation of marine bacteria, and it may also modulate the interaction between these bacteria and their symbiotic hosts. Coral-related bacteria can produce secondary metabolites [52] and metabolize DMSP [15]. The two cold-water corals in this study are soft corals living in extreme marine environments, and they should have evolved special adaptation mechanisms compared to Scleractinia corals, including the metabolism of their symbiotic bacteria [13].

#### 2.5.1. Secondary Metabolite Biosynthetic Gene Clusters

Since more than 200 novel chemical structures have been described annually [53], corals and their associated microbial communities have been considered prolific reservoirs of bioactive natural products. Many natural products obtained from the coral holobiont possess potent antibacterial, antiviral, anticancer, anti-inflammatory, antimalarial, and neuroprotective properties [54,55] and are thus of enormous potential for the blue economy sector [56]. Gorgonian coral *Eunicella labiate*-derived *Sulfitobacter* sp. EL44 is able to inhibit the growth of four fungal pathogens of the *Candida* genus and possesses several secondary metabolite biosynthetic gene clusters (SM-BGCs), such as ectoine, T1PKS (type I polyketide synthases), and hserlactone BGCs [57]. To predict the SM-BGCs of *S. faviae* SCSIO W1865 and SCSIO W1865, the whole genome sequences were uploaded to an-tiSMASH 7 beta. *S. faviae* SCSIO W1865 and SCSIO W1865 possessed at least seven and six putative SM-BGCs, respectively (Figure 6). Both of them harbor an ectoine cluster that may help microorganisms survive a myriad of environmental stresses by involving them in osmoregulation [58] and a T1PKS (type I polyketide synthases)-NRPS (non-ribosomal peptide synthase)-like hybrid cluster, which is often detected in bacteria and fungi and is involved in the biosynthesis of oligopeptides and polyketides [59]. β-lactone BGCs (two in SCSIO W1865) and hserlactone BGCs (two in SCSIO W1866) were found in the genomes of both strains. In addition to the above shared polyketide synthesis-related cluster, one terpene, SM-BGC, which may also serve as a chemical defense to protect coral from predators [60], was specifically found in SCSIO W1865.

The two strains shared five putative SM-BGCs, but specific SM-BGCs were also identified. This finding suggests that these two strains possess the potential to produce at least five different types of natural products with distinct skeletons. Interestingly, Cluster 3 and Cluster 6 were carried by SCSIO W1865 Chr. and Pla.1 separately. Bacteria can gain an advantage over competing microorganisms by exchanging secondary metabolite BGCs through horizontal gene transfer events, similar to plasmid-mediated delivery of antibiotic resistance [61]. β-lactone SM-BGCs carried by SCSIO W1865 Pla.1 can be exchanged among marine bacteria and may confer a selective advantage on the accepted strains when SCSI W865 encounters other strains. Clusters 5 and 11 showed 100% similarity with ectoine (Appendix A). Cluster 3 only exhibited 13% identity with the known SM-BGCs of corynecins, and the remaining 8 clusters had no similarity with previously reported SM-BGCs, and they are the potential sources for genome mining.

#### 2.5.2. Sulfite Oxidation and DMSP Degradation Pathways

*Sulfitobacter faviae* can gain energy from sulfite oxidation [2,9]. To investigate whether SCSIO W1865 and SCSIO W1866 have the ability to oxidize reduced sulfur compounds, we searched for the enzymes involved in the thiosulfate oxidation process mediated by the SOX complex. As shown in Appendix A, orthologs of genes that encode the thiosulfate-oxidizing sulfur oxidizing enzyme (Sox) system *soxXYZABCDGH* were identified in both strains, and they shared high similarities (95.12–100%). Based on their characteristics, SoxAX, SoxYZ, and SoxCD form complexes, and their encoding genes are usually clustered with other SOX complex-related genes such as *soxB*, *soxG*, *soxH*, *soxW*, *soxV*, *soxS*, *soxR*, *YeeV*. In the SOX complex, SoxAX is a heterodimeric c-type cytochrome mediating electron transfer; *soxB* is responsible for hydrolyzing cysteinyl S-thiosulfonate to cysteinyl persulfide and sulfate; SoxCD encodes the essential sulfur dehydrogenase of the reaction mechanism; and SoxYZ binds to substrate [62]. In both strains, the genes (except for *soxG* and *soxH*) are located between the Clp protease gene *clpP* and the putative phosphatase gene, while the thiosulfate-induced periplasmic zinc metallohydrolases *soxG* and *soxH* are neighbor genes located between a hypothetical protein and a membrane protein (FIG137887), suggesting that they both harbor enzymes in sulfite oxidation, which offer energy to the survival of the two strains.

The synthesis and degradation pathways of DMSP were identified and summarized in marine bacteria [10,63,64]. DsyB is an important methyltransferase enzyme responsible for the synthesis of DMSP, probably using the same methionine (Met) transamination pathway as macroalgae and phytoplankton [65]. DMSP production and *dsyB* transcription are upregulated by increased salinity, nitrogen limitation, and lower temperatures [10]. The *dsyB* gene was identified in coral-associated Alphaproteobacteria [66], but homologs of *dsyB* were not identified in the two cold-water coral-derived *Sulfitobacter faviae* strains. We further explored the potential DMSP degradation pathways in the two strains (Figure 7 and Appendix A). In the demethylation pathway, four genes, *dmdA* and *dmdBCD*, that are responsible for degrading DMSP to acetaldehyde and methanethiol were all identified [67]. However, the two lysis pathways seemed incomplete. In the pathway involving DddP and AcuK to lyse DMSP to DMS and 3-HP, only homologs of DddP were identified in both strains. In the second lysis pathway involving DddD to lyse DMSP to 3-HP-CoA and DddAC to convert HP-CoA to acetyl-CoA, only homologs of DddAC were identified in both strains. Thus, it seems that these two strains can degrade DMSP via the methylation pathway and can also convert DMSP to DMS via DddP. Whether these enzymes are involved in the interaction of *Sulfitobacter* with coral remains to be explored.

## 3. Experimental Procedures

### 3.1. Isolation of SCSIO W1865 and SCSIO W1866

Cold-water corals *Dendrobathypathes* sp. and *Telopathes* sp. were collected from two sites at 115.3945° E, 14.0321° N, and 115°0.684′ E, 13°20.568′ N at depths of ~1000 m during a South China Sea Open Cruise in September 2020, and the temperature of sea water is 0–4 °C. After cleaning with sterile sea water, the tissues were homogenized with a refiner (Tiangen, Beijing, China). The obtained homogenates were streaked on Marine Agar 2216E (BD Difco) plates with 1.5% agar. After culture at 4 °C for 7 days, single colonies were selected and cultured in 2216 E medium for two days at 25 °C. Cells for TEM were prefixed in 3% (*w*/*v*) glutaraldehyde at room temperature for five hours and dehydrated with increasing concentrations of ethanol. Then the cells were washed twice with tert-Butanol and frozen dry. Ultrathin (60-nm thick) sections of dry samples were cut and then mounted on a copper grid and observed via transmission electron microscopy (TEM, Čatež, Slovenia, Hitachi S-3000N).

The taxonomic classification of the single colonies obtained was determined by sequencing the 16S rRNA gene. Primer pairs F27 (5’-AGAGTTTGATCCTGGCTCAG-3′) and R1492 (5’-GGTTACCTTGTTACGACTT-3′) [68] were used to amplify 16S rDNA by polymerase chain reaction (PCR), and total genomic DNA was used as a template. Genomic DNA was isolated from cell pellets of the two *Sulfitobacter faviae* strains SCSIO W1865 and SCSIO W1866 with a Bacterial DNA extraction Kit (Tiangen) according to the manufacturer’s instructions. The PCR master mix Primer STAR Max (Takara, Japan) was used for PCR, and the PCR cycling conditions were 95 °C for 10 min, 40 cycles 95 °C/30 s, 60 °C/30 s, 72 °C/1 min, and the reaction was extended at 72 °C for 10 min. The obtained PCR products were purified using a gel extraction kit (Omega Bio-tek Inc., Norcross, GA, USA), and the obtained PCR products were purified and sequenced with the above primer pairs by Tianyi Huiyuan Biotech Co., Ltd. (Beijing, China) using the Sanger platform. The DNA sequence data were analyzed using Sequencing Analysis 5.2, and the DNA sequence data were analyzed using BLAST in NCBI.

### 3.2. DNA Extraction, Genome Sequencing, Assembly and Annotation

Genomic DNA isolated as above was quantified using a TBS-380 fluorometer (Turner BioSystems Inc., Sunnyvale, CA, USA). The quality of DNA was determined using NanoDrop™ 2000 Spectrophotometer (Thermo Fisher Scientific, Waltham, MA, USA), and Genomic DNA was quantified using the TBS-380 Fluorometer (Turner BioSystems Inc., Sunnyvale, CA, USA). High-quality DNA (OD260/280 = 1.8~2.0, >6 μg) was utilized to construct fragment libraries. A combination of Pacific Biosciences (PacBio) and Illumina sequencing platforms was used to sequence the genomes of the two strains at Shanghai Biozeron Biotechnology Co., Ltd., Shanghai, China. The Illumina data were used to evaluate the complexity of the genome and correct the PacBio long reads. The genomes were assembled with ABySS [69] and canu [70]. Then, GapCloser software was subsequently applied to fill the remaining local inner gaps and correct the single base polymorphism for the final assembled sequences [71]. The genomes were annotated with GeneMark [72], the NCBI nonredundant database [73], SwissProt [74], KEGG [75], COG [76], tRNAscan-SE [77], and RNAmmer [78].

### 3.3. Comparative Genomic Analysis

The comparative genomic analysis was conducted using cd-hit (v4.6.1) [79] with parameters of identity > 50% and coverage > 50%. All other parameters were used in the default settings.

### 3.4. Phylogenetic Tree Construction

The phylogenetic tree based on the sequences of 16S rRNA from genomes was constructed by using the MEGA 7 [80] maximum-likelihood algorithm with 1000 bootstraps after alignment by MUSCLE, where the Kimura two-parameter model [81] was employed. The initial tree for the heuristic search was obtained automatically by applying the Neighbor-Join and BioNJ algorithms to a matrix of pairwise distances estimated using the Maximum Composite Likelihood (MCL) approach and then selecting the topology with the superior log likelihood value. Sequences of 16S rRNA genes from the closely related genus *Jannaschia* were used as the outgroup. Branch lengths are proportional to the number of nucleotide substitutions.

The maximum likelihood tree based on the whole-genome sequences of SCSIO W1865, SCSIO W1866, and the available *Sulfitobacter* genomes listed in Appendix A was constructed using PhyML 3.0 software [82]. All genomes were aligned against the reference genome SCSIO W1865 using MUMmer (v3.0) [83] to generate whole-genome alignments and to identify single-nucleotide polymorphisms (SNPs) in the core genome, with repetitive regions removed. In total, 10,636 SNPs were identified in these 26 genomes. Based on the concatenated SNPs, a maximum-likelihood tree with 1000 fast bootstrap replicates was inferred using PhyML 3.0 under the GTR + I + G substitution model. Branch lengths are proportional to the number of nucleotide substitutions.

### 3.5. FastANI and Pacbio Genome Assembly of Sulfitobacter faivae S5-53^T^

The raw genome sequencing data of *Sulfitobacter faivae* S5-53^T^ were downloaded from NCBI (accession number: SRR6128330) and assembled by canu [70] with the tool Circulator [84] to cyclize the genome. All the parameters were used in the default settings. Pairwise ANI calculations between all collected Sulfitobacter genus genomes listed in Appendix A were performed using the tool fastANI v. 1.33 [85]. The alignment options were as follows: minimum length of 700 bp, minimum identity of 70%, minimum alignment of 50%, BLAST window size of 1000 bp, and step size of 200 bp.

### 3.6. Prediction of Metabolism

The secondary metabolite biosynthetic gene clusters were predicted in SCSIO W1865 and SCSIO W1865 using the antiSMASH 7 beta online program [86]. The genes involved in metabolic pathways of dimethylsulfoniopropionate (DMSP) degradation were identified by BLASTing the known proteins [64] against the annotated proteins in both SCSIO W1865 and SCSIO W1866.

### 3.7. Prediction of MGEs

IslandViewer 4 [28] was used to identify GIs (genomic islands). The prophages encoded by SCSIO W1865 and SCSIO W1866 were predicted by a combination of the phage search tool (PHASTER) [25] and Prophage Tracer [87]. A direct submission of the genome sequences was performed for PHASTER, and raw data from Illumina HiSeq sequencing of genomes was used for Prophage Tracer with default parameters.

### 3.8. Prediction of Antiphage Defense Systems

The antiphage defense systems were predicted with both DefenseFinder and the prokaryotic antiviral defense locator (PADLOC) [88,89]. The toxin-antitoxin systems were predicted by a combination of online rapid annotation using subsystem technology (RAST) 2.0 [90] and the TADB online prediction database [91]. CRISPR–Cas was predicted by the online software CRISPRone [92]. Genome sequences were submitted to the above web servers and analyzed with default parameters.

### 3.9. Data Availability

The whole genomes of SCSIO W1865 and SCSIO W1866 were deposited in NCBI’s GenBank Database under BioProject PRJNA923764, and the BioSamples are SAMN32735605 and SAMN32735606. The accession numbers of chromosomes and megaplasmids are listed in Table 1.

## 4. Conclusions

In summary, the whole genome sequences of two *Sulfitobacter faviae* strains derived from deep-sea cold-water corals were obtained. Bioinformatics analysis showed that these two closely related strains shared two highly similar plasmids and one prophage but also harbored many distinct MGEs. These MGEs enriched the genetic diversity of the two *Sulfitobacter* strains, likely increasing host fitness in the deep sea via carrying cold-adaptation-related regulatory modules and enzymes. Further investigations are warranted to explore the role of *Sulfitobacter* in the coral holobioint and the function of MGEs in mediating the interaction between *Sulfitobacter* and cold-water coral hosts.

## Figures and Tables

**Figure 1 marinedrugs-21-00309-f001:**
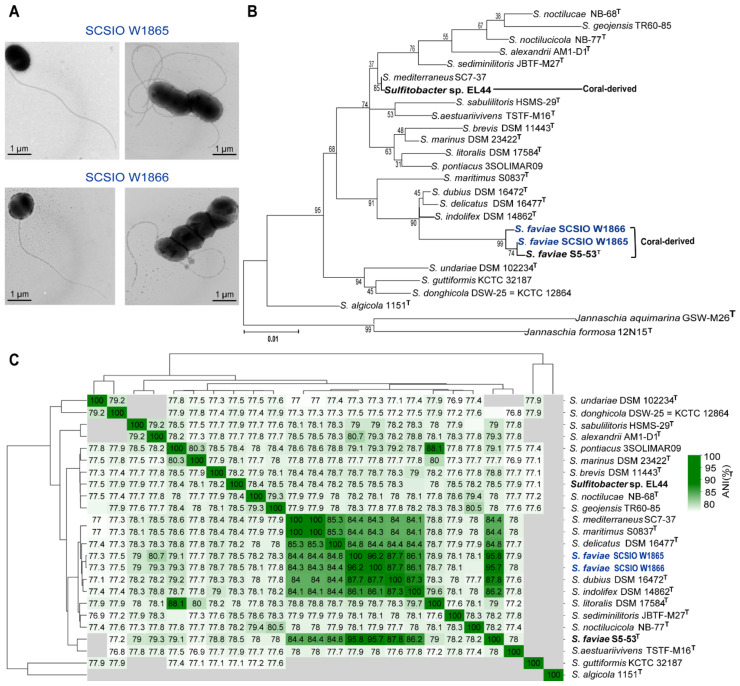
**Two *Sulfitobacter* strains were isolated from the cold-water black corals.** (**A**) TEM images of SCSIO W1865 and SCSIO W1866 cultured in 2216E medium. (**B**) The maximum likelihood tree based on the sequences of 16S rRNA genes of SCSIO W1865, SCSIO W1866, and the available *Sulfitobacter* listed in Appendix A. (**C**) ANI of all collected *Sulfitobacter* genus strain genomes listed in Appendix A. No ANI value is reported for a genome pair if the ANI value is much below 70%.

**Figure 2 marinedrugs-21-00309-f002:**
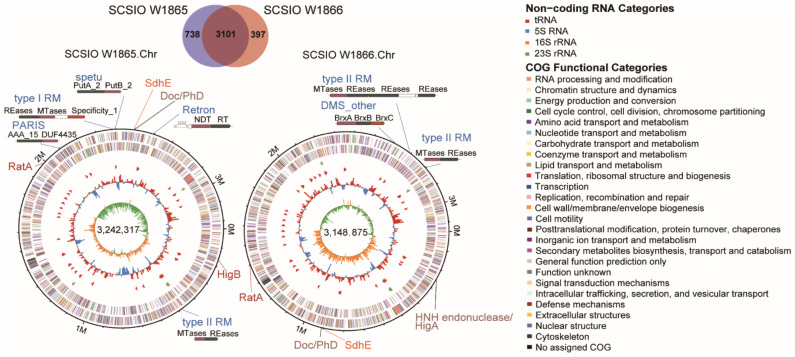
**Comparative genomic analysis of SCSIO W1865 and SCSIO W1866.** The Venn diagram illustrates the core genes and strain-specific genes between SCSIO W1865 and SCSIO W1866 (upper panel). The annotation of the chromosomes of SCSIO W1865 and SCSIO W1866 is also shown (lower panel). The seven rings from outermost to innermost indicate scale marks of the genome: protein-coding genes on the forward strand, protein-coding genes on the reverse strand, rRNA, tRNA, GC content, and GC skew. The putative toxin-antitoxin (TA) pairs are shown in brown letters, while the orphan toxins and antitoxins are shown in red and orange letters, respectively. The other antiphage systems are shown in blue letters.

**Figure 3 marinedrugs-21-00309-f003:**
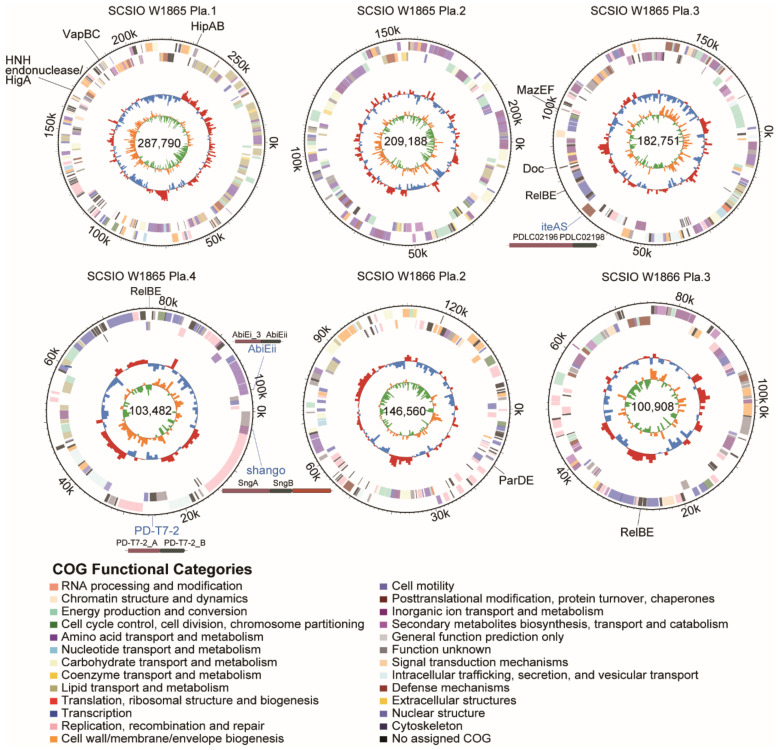
**Two strains harbor several megaplamids that carry diverse antiphage elements.** The complete genomes of four plasmids in SCSIO W1865 and two plasmids in SCSIO W1866 are shown. SCSIO W1865 Pla.2 and SCSIO W1866 Pla.1 showed 96.2% identity, and only the former one is shown here. The five rings from outermost to innermost indicate scale marks of the genome, protein-coding genes on the forward strand, protein-coding genes on the reverse strand, GC content, and GC skew. The TA genes are shown in black, and other types of antiphage systems are shown in blue.

**Figure 4 marinedrugs-21-00309-f004:**
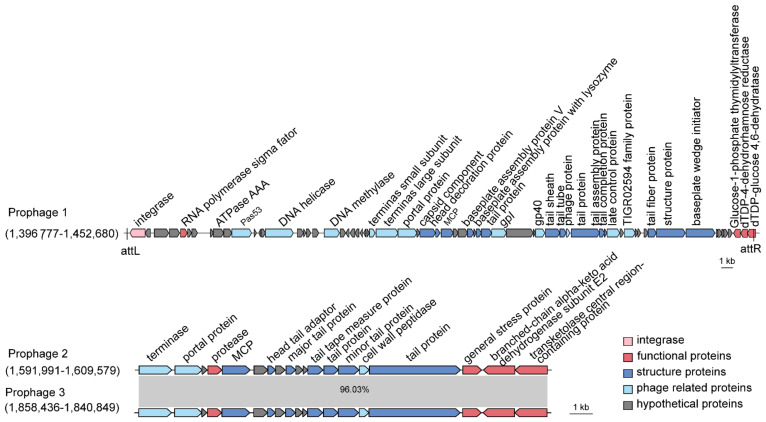
**SCSIO W1865 and SCSIO W1866 harbor prophages.** Prophages 1 and 2 are carried by SCSIO W1865, and prophage 3 is carried by SCSIO W1866. The function of prophage-encoded genes is shown at scale. The genomes of prophage 2 and prophage 3 were aligned using NCBI BLAST. MCP: major capsid protein.

**Figure 5 marinedrugs-21-00309-f005:**
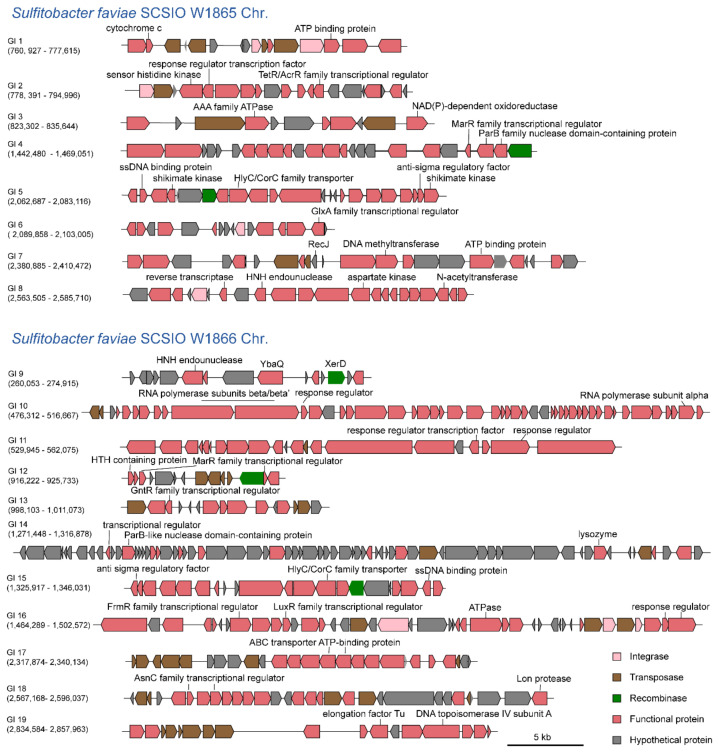
**SCSIO W1865 and SCSIO W1866 harbor GIs.** GIs in SCSIO W1865 and SCSIO W1866 chromosomes were predicted using IslandViewer 4. Those that were predicted by at least one method integrated by IslandViewer 4 were selected. The numbers below GIs indicate the positions of the corresponding GIs in the chromosome.

**Figure 6 marinedrugs-21-00309-f006:**
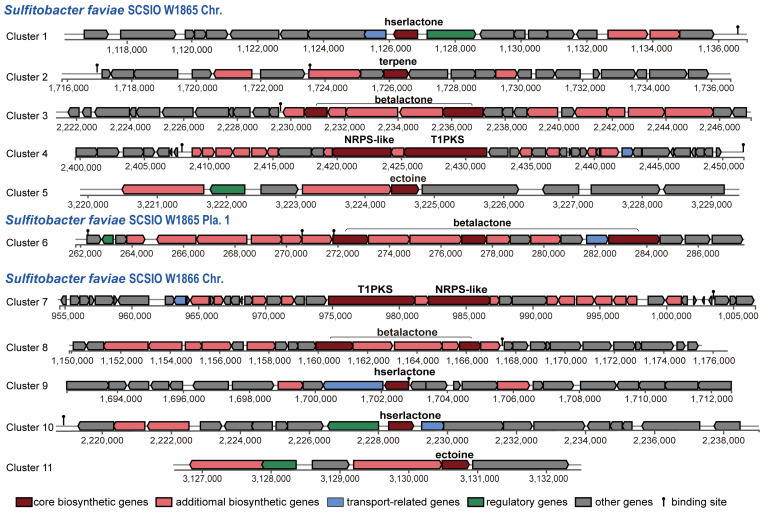
Modular organization of secondary metabolite gene clusters from SCSIO W1865 and SCSIO W1866.

**Figure 7 marinedrugs-21-00309-f007:**
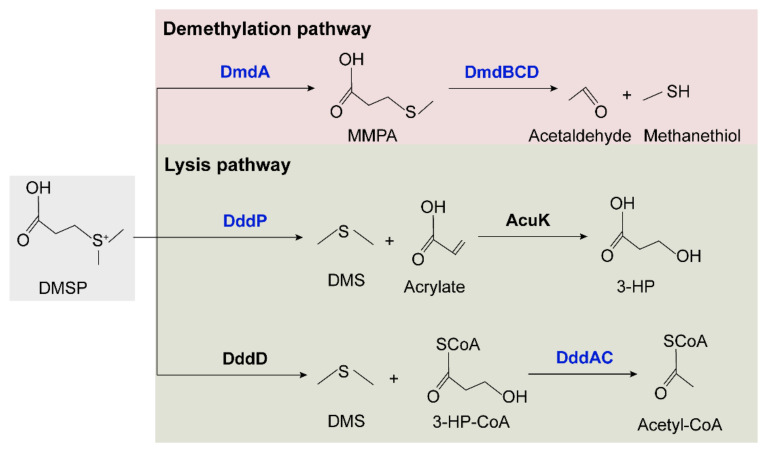
**Proposed DMSP degradation pathways in SCSIO W1865 and SCSIO W1866.** The enzymes identified in the two strains are shown in blue, and the DddD and AcuK enzymes that were not identified in the two strains are shown in black.

**Table 1 marinedrugs-21-00309-t001:** General genomic information of SCSIO W1865 and SCSIO W1866. Chr. and Pla. indicate chromosome and endogenous megaplasmid, respectively.

Features	SCSIO W1865					SCSIO W1866			
	Chr.	Pla.1	Pla. 2	Pla. 3	Pla. 4	Chr.	Pla.1	Pla. 2	Pla. 3
Topology	circular	circular	circular	circular	circular	circular	circular	circular	circular
Assembly sizes (bp)	3,242,317	287,790	209,188	182,751	103,482	3,148,875	210,697	146,560	100,908
G + C content (%)	62.72	57.55	63.53	61.03	59.88	62.76	63.54	57.27	59.21
Protein coding genes	3185	276	192	188	98	3122	195	152	116
tRNA genes	49	0	0	0	0	49	0	0	0
rRNA genes	12	0	0	0	0	12	0	0	0
Accession number	CP116423	CP1164224	CP116425	CP116426	CP116427	CP116419	CP116420	CP116421	CP116422

**Table 2 marinedrugs-21-00309-t002:** TA systems were predicted in SCSIO W1865 and SCSIO W1866 by RAST and TADB. The identical TA pairs found in the two strains are shown in bold. N/A indicates that it is not applicable.

Name	Classification	Origin	Toxins	Start	Stop	Antitoxins	Start	Stop	Strand
**TA systems**									
Doc/PhD	type II	Chr.W1865	Doc	2,486,830	2,487,207	PhD	2,486,585	2,486,833	+
Doc/PhD	type II	Chr.W1866	Doc	911,438	911,061	PhD	911,683	911,435	−
HipAB	type II	Pla.1 W1865	HipA	231,854	233,173	HipB	231,603	231,854	+
ParDE	type II	Pla.2 W1866	ParE	123,64	12,672	ParD	12,126	12,377	+
**RelBE**	**type II**	**Pla.3 W1865**	**RelE**	**77,137**	**76,862**	**RelB**	**77,423**	**77,124**	**−**
**RelBE**	**type II**	**Pla.3 W1866**	**RelE**	**27,843**	**28,118**	**RelB**	**27,557**	**27,856**	**+**
RelBE	type II	Pla.4 W1865	RelE	77,561	77,220	RelB	77,223	76,930	−
MazEF	type II	Pla.3 W1865	MazF	100,842	101,237	MazE	100,625	100,852	+
VapBC	type II	Pla.1 W1865	VapC	186,308	186,532	VapB	185,844	186,107	+
HNH/HigA	type II	Pla.1 W1865	HNH endonuclease	168,456	167,224	HigA	169,069	168,449	−
HNH/HigA	type II	Chr.W1866	HNH endonuclease	264,680	263,349	HigA	264,965	264,690	−
**Orphan toxins**									
Doc	type II	Pla.3 W1865	Doc	83,972	83,811	N/A	N/A	N/A	−
HigB	type II	Chr.W1865	HigB	163,404	162,178	N/A	N/A	N/A	−
**RatA**	**type II**	**Chr.W1865**	**RatA**	**2,005,676**	**2,005,224**	**N/A**	**N/A**	**N/A**	**−**
**RatA**	**typeII**	**Chr.W1866**	**RatA**	**1,402,847**	**1,403,299**	**N/A**	**N/A**	**N/A**	**+**
**Orphan antitoxins**									
SdhE	type II	Chr.W1865	N/A	N/A	N/A	SdhE	2,482,046	2,481,780	−
SdhE	type II	Chr.W1866	N/A	N/A	N/A	SdhE	916,222	916,488	+

**Table 3 marinedrugs-21-00309-t003:** Loci and gene structure of predicted antiphage systems.

Loci	Antiphage Systems	Start	Stop	Strand
SCSIO W1865 Chr.	retron 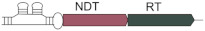	msr-msd	2,571,487	2,571,320	−
NDT	2,571,402	2,570,368	−
RT	2,570,375	2,569,410	−
type I RM 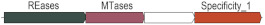	REases	2,394,724	2,397,186	+
MTases	2,397,190	2,398,638	+
type II RM 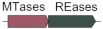	MTases	476,565	475,996	−
REases	476,012	475,281	−
PARIS 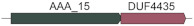	AAA_15	2,122,422	2,121,121	−
DUF4435	2,121,124	2,120,267	−
spetu 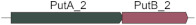	PtuA_2	2,403,386	2,404,963	+
PtuB_2	2,404,967	2,405,818	+
SCSIO W1865 Pla.3	iteAS 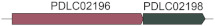	PDLC02196	66,640	67,722	+
PDLC02198	67,775	70,093	+
SCSIO W1865 Pla.4	Shango 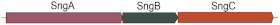	SngA	1	2544	+
SngB	2522	3829	+
SngC	3835	6012	+
AbiEii 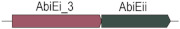	AbiEi_3	102,238	101,408	−
AbiEii	101,418	100,612	−
PD-T7-2 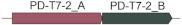	PD-T7-2_A	25,827	24,583	−
PD-T7-2_B	24,586	22,769	−
SCSIO W1866 Chr.	type II RM 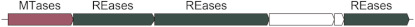	MTases	271,000	269,702	−
REases	269,691	268,174	−
REases	268,177	265,946	−
REases	264,680	263,349	−
type II RM 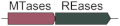	MTases	2,822,081	2,822,650	+
REases	2,822,634	2,823,365	+
DMS_other 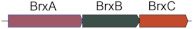	BrxA	2,589,408	2,588,800	−
BrxB	2,588,798	2,588,196	−
BrxC	2,588,192	2,584,668	−

## Data Availability

The authors declare that all relevant data supporting the findings of this study are available within the article and its Appendix A files. The whole genomes of both SCSIO W1865 amd SCSIO W1866 were deposited in NCBI database with accession numbers CP116423–CP116427 and CP116419–CP116422.

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
