# Peer review of "Comparative Genomic Analysis of Cold-Water Coral-Derived Sulfitobacter faviae: Insights into Their Habitat Adaptation and Metabolism"

_marinedrugs, 2023, doi:10.3390/md21050309_

Round 1
Reviewer 1 Report (New Reviewer)
This is a well written manuscript that describes a competent comparison of certain characteristics of the genomes of two bacterial strains isolated from cold water corals. The methods and results are adequate to support the conclusions. That being said, the conclusions are fairly unremarkable. Nothing about these analyses gives anything more than the most general idea of how these bacteria interact with their coral hosts, or their environment, or gives much insight into what unique compounds they may produce. Deeper investigation of these questions might add more interest to this study. Nonetheless, I can find no reason not to recommend publication. I have attached a marked-up copy of the manuscript that includes minor corrections and suggestions.

Author Response
Please see the attachment.

Reviewer 2 Report (New Reviewer)
Review of the manuscript entitled Comparative genomic analysis of cold-water coral derived Sulfitobacter faviae: insights into their habitat adaptation and metabolism by Lin and colleagues submitted to Marine Drugs. The introduction is sound, the experimental procedures are well described but it needs some improvements and the results and discussion well explored the results and discussion is restricted, not because the authors did not search the literature, but the information is actually limited. I have a few comments, questions and suggestions that I hope will improve the manuscript.
Introduction
L. 34: Please fix the name of the bacteria.
L. 39-40: With theoretically around 27 species according to the LPSN, there are some members that are “mostly are catalase- and oxidase-positive” and other catalase- and oxidase-negative?
L. 42-43: Including Sulfitobacter faviae from the coral Favia veroni, right?
L. 43-45: In corals in general or in deep-water corals?
L. 47: Please, fix the “organic”.
L. 49: “reef-building coral Acropora tenuis” at what depth?
L. 56-58: I know it is not the scope of the ms, but do the authors know whether or not Archaea also play a role regarding the DMSP?
L. 64-65: and depth as well? How about hosts, is it also holds true?
L. 66: There is a missing space between “(MGE)” and “are”.
L. 73-74: Is there an estimation regarding the number of phage or virus that can be found at the deep-sea?
Results and discussion
L. 80: I know the focus here is the isolation of Sulfitobacter, but the first topic in MM described the isolation procedure. Thus, I would like to know how many isolates were obtained, which percentage corresponded both Sulfitobacter strains? How where their colonies in terms of shape, colour, and son on? The isolates’ sequences were submitted to the NCBI? which are their accession numbers?
L. 83: How TEM was performed? Please, explain the procedure in the appropriate section.
L. 88-89: It is something that needs to be clarified at MM that 16S rRNA gene from the sequencing amplification and genomes were used. It took me a while to figure out that two inferences were performed, one for 16S rRNA gene sequences and another one from the genomes. Please, add a topic to exclusively explain how the phylogenetic inferences were performed (data set, models, and so on), it will definitively, assist the readers to understand what was done.
L. 90-93: I believe a “respectively” is missing in this phrase.
L. 98: Why neighbour joining was used? There are more robust phylogenetic inferences, such as maximum likelihood? Since the tree was done at MEGA, how about the most appropriate model for the data, was It verified?
Legend figure 1: The details in item B should be provided at the MM section.
L. 100: In Table S1, if it is possible, please indicate the phylum each organism belongs to, as it was done “Gorgonian coral Eunicella labiate”
Table 1: In features, please, add “number” after “accession”
L. 122-123: Which other MGEs would be? Please, clarify.
L. 138-140: It was already stated above and it is present in table 1, so summarize it or remove this information form the above topic.
L. 141-143. Here it is stated again how many megaplasmids are in each strain.
L. 145-147: In “metabolic process (phosphorus metabolic process, carbohydrate metabolic process, cellular nitrogen compound metabolic process and urea metabolic process),” It is clear that the categories between brackets belong to metabolic process, is there a way to avoid repeating it after every single process?
L. 147-149: Same here: “response to stimulation (stress response to metal ion, response to osmotic stress and response to oxidative stress),” suggestion “(stress response to metal ion, osmotic stress and oxidative stress)”
L. 148-149: “and some other pathways especially those involved in biological processes” such as???? Ok, I can see that several examples are provided in Fig. S3 but please, give 2-3 examples here.
L. 149: Figure S3: Provide the reference for eggnog software, please.
L. 149-150: Where it was described? Fig. S3? Fig. 3?
Figure 3: Why one plasmid from SCSIO W1866 was not showing? Perhaps because it is very similar to ACSIO W 1865 Pla2? If so, make it clearer, please.
L. 174-176: Is that is the reason why it is not possible to know how these prophages would affect both Sulfitobacter faviae?
L. 178: Is 50% similarity enough to attribute the prophages to a virus group? How members of the Myoviridae phages interact with the host?
L. 178-185: How it would affect the host bacteria? Specially prophages 2 and 3? How will it affect the host coral?
L. 189-194: Prophages 2 and 3 seem to play a very important rule not only for the bacterium host but also for the coral, especially considering the environment conditions that a deep-sea coral deals every single day!
L. 176-177: MCP, which is the correct name: major coat proteins or L. 199: major capsid protein. I believe the acronym has to have the same name.
L. 210-202: In the same coral host?
L. 203: “eight” instead of “8”, please, fix it.
L. 204: Please, specify that in SCSIO W1866, there were 10 distinct GIs.
L. 204-211: I was wondering if a table with this information will be more appropriate than the text.
Figure 5: What are the number below the GI #, for instance, GI 1 (760,927 – 777,615)? The position of the GI in the chromosome?
L. 226: I would suggest to keep only “microbiome”.
L. 231-233: The Type II TAs are the most common, because it is the most studied so far?
L. 233-235: Does it mean that the other TAs Types were not detected or the focus was on Type II TAs?
L. 235-247: All those systems belong to Type II TAs, right?
L. 251-253: Was this true for all the systems described above?
L. 253-255: Do the authors know how often it does happen?
Table 2: Why the third RelBE in the table was not marked? Because it happens in only one strain, so there is no way to know whether or not they are identical?
L. 266-267: However, according to Table 1 both stains have only one chromosome. Perhaps what the authors meant was “were distributed in both, the chromosome and megaplasmids 3 and 4?
L. 269-270: I did not understand this phrase, “most” seems to be misplaced here.
L. 271-274: It means that they are not the same as it was shown for the RelBE, right? But even then, the RM systems still be able to perform the same function, is that what the authors meant?
L. 277-278: But even with the possibility to exchange between strains, they were not found in SCSIO W1866
L. 281-282: Please change “5” to “five” and “4” to “four”.
L. 262-287: All these types of antiphage systems have the functionality?
L. 299-301: These activities were obtained from strains or predicted from the genomes?
L. 308: “Both of them harbour” means both strains, right? Make it cleared, because in the previous phrase there was the “at least seven and six putative SM-BGCs, respectively”.
L. 309: Please shift “survive myriad environmental stresses” to “survive a myriad of environmental stresses
L. 312-314: I did not understand, if beta-lactone and hserlactone were deleted in both strains, why it seems that the first one was only detected in W1865 and the latter in W1866?
L. 314-315: But polyketides can also produce compounds with several activities, among them, against predator.
L. 325: It is a very low identity, corynecins is probable the closest match, but I am ascertaining that it is not corynecins, right?
L. 316-317: And most likely new compounds or different structure of known compounds.
L. 329: Please indicate in the W1866 that those cluster were detected in the chromosome, please.
L. 335-L357: Please, change the order of the Supplementary table, such as Table S4 is actually S3 and consequently, S3 is in reality S4.
Experimental procedures
L. 385: Is 2216E the marine medium? Please, provide the manufacture.
L. 386-387: The colonies did not take that long to growth. I would have imagined that the cultivation would take weeks to occur considering the differences in light, nutrients, and so on. I am also surprised that the isolates grew in two days at 25C.
L. 388-395: How DNA was extracted? Please, provide the references for the primer pair. Move the primer´s sequence next to the its name, so there is no need to repeat the primer´s name. Change 16S rDNA to 16S rRNA gene. Which master mix for PCR was used? How the amplicons were purified? Both primers were used for the sequencing? Which sequencing platform was used, Sanger? I am sure that BLAST was not used to analyse the sequences, instead, it was used to identify the sequences. Which software was used to analyse the sequences. Were the sequences submitted to any data bank, such as the NCBI? If so, which are the accession numbers?
L. 399: How DNA quality was determined?
L. 405-407: Provide the references for both softwares, please.
L. 412: Which 16S rRNA gene was used, the ones obtained in the above topic or from the genomes, please clarify?
L. 417-421: Please, fix the name of the pacbio. Italicized the bacterium’s name, please.
L. 427: Provide the references of the antiSMASH.
L. 432: “GIs” stands for?
Author Response
Please see the attachment.

Reviewer 3 Report (New Reviewer)
Please see the comments in the manuscript provided.

Round 2
Reviewer 3 Report (New Reviewer)
No further comment.
This manuscript is a resubmission of an earlier submission. The following is a list of the peer review reports and author responses from that submission.
Round 1
Reviewer 1 Report
In the manuscript, the authors describe the functional analysis, and corresponding comparison, of the genome of two Sulfitobacter faviae strains isolated from cold-water corals. In their analysis, they mainly focused on the identification of the various Mobile Genetic Elements (MGEs), antiphage systems, secondary metabolite biosynthetic gene clusters, and pathways for the degradation of dimethylsulfoniopropionate (DMSP).
Their findings shed light on the role of MGSs in this bacterial genus, how it is able to counteract phages, and how they contribute to coral-bacterial interactions that is crucial for sustaining coral health.
The article flows easily, is well organised, with appropriate figures and tables. However, I found some errors, which can be easily resolved, and some sentences that are not clear to me. In this regard, please check the attached file.

Reviewer 2 Report
The manuscript is very well written and present interesting scientific results. The authors compare the genome of two Sulfitobacter strains isolated from different cold-water corals. The authors suggest insights into the adaptive strategy of Sulfitobacter strains to thrive in ecological niches. Although the work is good, the mains goal is an ecological investigation, which does not fits in the scope of this journal. The authors described an analysis of the biosynthetic gene cluster, however, they were able to identify only a BGC of a osmolyte metabolite with 100%, and a possible antibiotic with 13% identity which means too low confidence. I suggest the authors should submit the manuscript to a more environmental related journal.
Author Response
The reviewer 2 did not raise specific comments to be addressed.
Reviewer 3 Report
I regret to say that I cannot recommend your manuscript for publication in Marine drugs.
The manuscript appears to me as a genome announcement paper, which includes the genomes of two closely related strains of Sulfitobacter faviae.
As much as I appreciate new complete genomes being added to the databases, and a short report describing them, I recommend the authors to look to other journals (than Marine drugs) that would be a better fit for this kind of report. A more specialized journal will likely accept it after some revision.
Here are a few recommendations for further improvements:
- Change (revise and shorten) the text into a "genome report" style based on the description from a journal (maybe the journal "Genome Announcements"?).
- Although the Introduction should be brief, as in this case, the current version jumps from one topic to the next many times, with only a few lines on each of them, so it becomes very confusing and not very informative.
- The figure 1 legend is incomplete. Although I recommend to replace the 16S rDNA tree with a new tree based on whole genomes or at least multiple genes, if possible. (Phylogenomics or MLSA approach)
-I am unsure how informative several of the figures are. e.g. Figs 4-6. Could perhaps be provided as supplements?
- The methods section is very minimalistic, and it would be difficult to replicate your work based on your text.
Best regards.
Reviewer 4 Report
General comments: This article explored the genomes of two cold-water coral-derived Sulfitobacter faviae strains with focus on mobile genetic elements, secondary metabolism, and DMSP pathways, since these can be key mediators of bacterial adaptation in marine biomes. Using an array of bioinformatic tools the authors did a comprehensive mining for plasmids, genomic islands, antiphage systems, and looked into secondary metabolite biosynthesis gene clusters and DMSP degradation potential. The study demonstrates that both Sulfitobacter strains possess several plasmids and prophages and describes the structure and gene content of these elements.
Main concerns:
Introduction: Firstly, the authors do not identify their research question, hypothesis, and/or objective to justify the work. Thus, the importance of the study is unclear. Upon revision, a statement of the study objectives must be added to the final paragraph of the introduction.
Secondly, it is not clear why these two Sulfitobacter flaviae strains/genomes are of particular interest and were chosen for this work and, if other Sulfitobacter strains and/or marine bacteria were co-isolated from these corals. This should be better contextualized in the paper. The authors mention in the introduction that Sulfitobacter is a sulfite-oxidizer, but they do not search for / explore this function on the genomes of their strains and do not discuss the role of sulfite oxidation for the deep sea coral holobiont. This should be improved.
Thirdly, the introduction lacks some introductory sentences about the coral species from which the Sulfitobacter strains were obtained, their taxonomic placement, natural habitat, ecology, and symbiotic status. Such insight is useful to contextualize this type of work and to justify its relevance.
Soundness of the Methodology. The manuscript does not seem to provide compelling enough evidence that the chromosome, megaplasmid and prophage sequences were complete and that all claimed plasmids are indeed plasmids. The authors should show quality parameters related with their assemblies, including contig and scaffold numbers, contamination and completeness scores etc and describe in detail how they inferred the number of plasmids and prophages in each strain. Did you confirm the presence of plasmids by a plasmid-specific DNA extraction and/or pulse-field gel electrophoresis? Such “wet-lab” proof would make the manuscript more solid and is very feasible for this small sample size (two bacterial strains).
Results and Discussion. Although the authors acknowledge in all results/discussion subsections that their findings are possibly involved in the adaptation and competitiveness of the bacterial isolates, the discussion is mostly shallow and does not dive into symbiont ecology and functioning. A better framing of the findings of this study within the research field and current literature should be conducted so that the reader can appreciate the value of this study.
Especially the part about the secondary metabolism / identification of BGCs is lacking depth, which, considering the scope and readership of Marine Drugs, should be improved.
Language. The English language needs to be improved, as does the writing style. There are several grammar and spelling mistakes and ambiguous or non-precise sentences in the manuscript. Especially the ‘Results and Discussion’ section contains multiple unclear sentences.
Figures. Resolution and font size of figures should be improved. Particularly in Figures 2 and 3.
Abbreviations. The manuscript contains many abbreviations that were not explained at their first mention in the text.
Specific comments:
Abstract:
Ls. 15/16. The sentence is a bit odd. Please rephrase. For example: “Their complex lifestyle and metabolism may have important ecological functions in association with eukaryotic hosts.”
L. 22. It should read “several”, not “serval”.
Ls. 25. Please rephrase the end of the sentence to clarify that you are studying genes. Add “...genes involved in...” before “dimethylsufoniopropionate”.
Keywords. The following keywords are redundant as they are also present in the title: cold-water coral, Sulfitobacter faviae, habitat adaptation. Please replace with more appropriate keywords.
Introduction:
L. 33. Insert space between “Rhodobacteraceae” and “[1]”.
L. 35/36. Why is it relevant here that most strains of the genus are catalase- and oxidase positive? I suggest removing this sentence, since it is not further related to / discussed in the paper.
L. 43/44. Avoid repeating similar sentences in your paper. The same phrase was present in the abstract. Substitute “contexts” with “roles” and rephrase sentence.
L. 49. I suggest rephrasing the start of the sentence into “ Stony and soft coral holobionts”. Since it is mostly the symbiotic algae that are producing the DMSP.
L. 50/51. The meaning of this sentence is unclear. Please rephrase to improve clarity.
L. 53. Substitute “for coral bleaching” with “to mitigate coral bleaching...”
L.54/55. Please specify the deep-water coral taxon/species.
L.58. It should read “habitat”, not “habit”. I suggest deleting the word “extreme”.
L.62. Please indicate the cold water coral species / genus, and the corals’ taxonomic placement (i.e., class Hexacorallia, order Anthipatharia (“Black corals”)) from which the Sulfitobacter strains were isolated.
L.63. Replace “Two” with “Both”.
L.63-68. These last four sentences of the introduction are a result summary, similar to what is said in the abstract. Replace this summary with the objectives of this study and explain to the reader your hypothesis or research question.
Results and Discussion:
Section 2.2 / L.89 onwards. How did you confirm that all the plasmids/megaplasmids are indeed plasmids and not contigs/scaffolds that were not assembled together with the chromosomes? Did you confirm this also experimentally, e.g., by plasmid-specific DNA extraction methods and/or pulse-field gel electrophoresis? Please clarify this in this paragraph.
L. 92. It should read “3.24 Mb”. The strain name in brackets behind is unnecessary and should be removed, as it is given at the end of the sentence.
L. 94. It should read “3.15 Mb”. The strain name in brackets behind is unnecessary and should be removed, as it is given at the end of the sentence.
L. 127. Please explain the abbreviation “GO pathways” at first mentioning in the text.
L. 127-130. Please be more detailed here and specify the type of metabolic processes, responses to stimuli (which?) and type of inter/intra-species interactions.
L. 133. Please correct the spelling: “megaplasmids”
L. 152. Specify which conditions of stress and add reference?
L. 154. Add reference for the PHAST program. Why did you use PHAST and not the upgraded version PHASTER which uses more up-to-date databases?
Ls.154-156. Please use past tense when reporting results. Check manuscript for consistency.
L. 165. Use past tense. Remove “they”.
L. 176. Insert “could” before “affect”. It is a possibility and you can speculate about it but, in fact, we lack experiment proof of it.
L. 187. “...may face strains containing diverse, mobile GIs...”. The meaning of this part of the sentence is unclear. Please reword and explain in more detail.
Subsection 2.4 Ls. 205-210. This section on antiphage systems is quite interesting and should be discussed in more detail. Indeed, high abundances of genes involved in antiviral defense mechanisms such as antiphage / CRISPR/CAS systems and also endonucleases was shown to be a feature of the healthy coral microbiome, see Keller-Costa et al., 2021 Microbiome (https://microbiomejournal.biomedcentral.com/articles/10.1186/s40168-021-01031-y) and of octocoral symbionts (https://link.springer.com/article/10.1186/s40168-022-01343-7) as well as of other prokaryote-marine invertebrate symbioses such as marine sponges (https://pubmed.ncbi.nlm.nih.gov/29312205/). The relevance of these features for holobiont ecology should be illuminated and literature in the field acknowledged.
Ls 212/213. Who is enriched in MGEs? Ambiguous wording, please rephrase sentence.
L.222. Correct spelling of the word “megaplasmids”.
Ls. 228-236. Please back up this sentence with a reference.
Ls. 232-234. It is not quite clear from your data that the incomplete TA systems present on the megaplasmids of your Sulfitobacter genomes are indeed a result of HGT. Can you provide further evidence from your own data for this? Has HGT of TA systems been reported in the literature? Please cite.
L. 251. Add “the” between “in” and “chromosome”.
L.260/261. Unclear. Do you mean “exchanged” among strains, instead of “delivered”? And how would this affect the physiological phenotype of the host? The connection is also unclear and needs to be explained better and backed up with references.
L. 269. Remove “with” after “hosts”. Consider replacing “hosts” with “bacteria”, to avoid confusion with the animal host.
L.275-278. This sentence makes little sense. The order Scleractinia (scleractinean corals) also contains cold-water / deep-sea corals (check the following study by Baco et al., 2017, Scientific Reports https://www.nature.com/articles/s41598-017-05492-w).
L.279. This section on BGCs is lacking a discussion and comparison with literature reports. For example, how do the BGC profiles of your Sulfitobacter strains compare to other strains isolated from coral (e.g. Sulfitobacter sp. EL44?) Check Almeida et al., 2022 Marine Drugs for the BGC profile and antimicrobial activities of strain EL44). Which roles do the identified BGCs have and how could they convey ecological advantages?
L. 282. Replace the word “tools” with the antiSMASH version used.
L.283/284. Please give full name of the BGCs hserlactone, T1PKS, NRPS etc.
L.287-289. Please improve clarity and wording of the sentence. Also, please substitute the word “delivered” with “exchanged”. Are BGCs /beta-lactones commonly found on plasmids and/or subjected to HGT. You could/should discuss.
L.308/309. This sentence seems incomplete. Please check and revise.
L. 309. Missing a space between “which” and “encodes”. It should read “lyses” not “lyse”.
L. 309/310. Sentence unclear. In the present study, you did not investigate gene expression.
Conclusion
L.324/325. This sentence needs rewording. For example: “These MGEs enriched the genetic diversity of the two Sulfitobacter strains, likely enhancing the fitness of these bacteria”.
L.325/326. This sentence needs revision. “The metabolism of secondary metabolites” does not make any sense. Specify which secondary metabolites and which are their putative functions (see my earlier comment).
L.327-330. This last conclusion sentence is a bit far fetched and stands out as not fitting in. The entire paper revolves around the ecology and habitat adaptation aspects of the two strains. Biotechnological aspects are neither explored in the Introduction, nor in the Result and Discussion section, wherefore I suggest removing this sentence.
Experimental procedures
L. 333. Specify which coral taxa.
Ls340/341. Substitute “the rDNA” with “the rRNA gene”.
Ls. 339-344. Did you perform a direct colony PCR or did you extract DNA first? Please specify. Please include PCR reaction conditions or a reference to the protocol used.
L. 347. Add “extraction” between “DNA” and “kit”.
L.350-355. Please specify how the genomes were quality checked and indicate genome metrics and quality parameters, such as number of contigs and scaffolds, completeness and contamination scores etc.
L.356-357. Please add versions and literature references to the annotation tools used.
Figures and Figure Legend:
Figure 1 Legend. Please state the tree-building method and evolutionary model used for tree construction.
Figures 2 and 3. Increase image resolution, font size, and overall figure size, which is too small.
Figure 2 and 3 Legend. Please explain each of the seven rings (from outermost to innermost ring) of the chromosome and plasmid maps in the figure legends.
Figure 3 Legend. (L.139) Please use the full name of the abbreviation "TA" the first time it appears in the text. Please consult the journals guide for authors about how to handle abbreviations correctly. Ls.141-143. Please double-check if the content of this sentence is correct and needed.
Figure 4 Legend. The word “structure” is spelled wrongly in the legend inside the figure.
Figure 5 Legend. Please use past tense (“were predicted”, not “are predicted”).
Table S1. Note that the genome of Sulfitobacter sp. strain EL44 was published as part of several papers: Raimundo et al., 2018 Marine Drugs (https://doi.org/10.3390/md16120485 ), Sweet et al., 2021 mSystems https://doi.org/10.1128/mSystems.01249-20; Almeida et al., 2022 Marine Drugs. Please add citation to the supplementary table.
Figure S1. Legend. Please replace “Comparation” with “Comparison”.
